# A multicenter explanatory survey of patients' and clinicians' perceptions of motivational factors in rehabilitation

Kazuaki Oyake [1,2], Katsuya Yamauchi[3], Seigo Inoue[2], Keita Sue[4], Hironobu Ota[5], Junichi Ikuta[6], Toshiki Ema[7], Tomohiko Ochiai[8], Makoto Hasui[9], Yuya Hirata[10], Ayaka Hida[11], Kenta Yamamoto[12], Yoshihiro Kawai[13], Kiyoto Shiba[14], Akihito Atsumi[15], Tetsuyuki Nagafusa[3] & Satoshi Tanaka [16 ✉]

## Abstract

**Background** Patient motivation is an important determinant of rehabilitation outcomes. Differences in patients' and clinicians' perceptions of motivational factors can potentially hinder patient-centered care. Therefore, we aimed to compare patients' and clinicians' perceptions of the most important factors in motivating patients for rehabilitation.

**Methods** This multicenter explanatory survey research was conducted from January to March 2022. In 13 hospitals with an intensive inpatient rehabilitation ward, 479 patients with neurological or orthopedic disorders undergoing inpatient rehabilitation and 401 clinicians, including physicians, physical therapists, occupational therapists, and speech-language-hearing therapists, were purposively selected using inclusion criteria. The participants were asked to choose the most important factor motivating patients for rehabilitation from a list of potential motivational factors.

**Results** Here we show that realization of recovery, goal setting, and practice related to the patient's experience and lifestyle are the three factors most frequently selected as most important by patients and clinicians. Only five factors are rated as most important by 5% of clinicians, whereas nine factors are selected by 5% of patients. Of these nine motivational factors, medical information ($p < 0.001$; phi $= -0.14$; 95% confidence interval $= -0.20$ to $-0.07$) and control of task difficulty ($p = 0.011$; phi $= -0.09$; 95% confidence interval $= -0.16$ to $-0.02$) are selected by a significantly higher proportion of patients than clinicians.

**Conclusions** These results suggest that when determining motivational strategies, rehabilitation clinicians should consider individual patient preferences in addition to using the core motivational factors supported by both parties.

## Plain language summary

Rehabilitation is the interventions needed to restore the abilities required for daily life following illness or injury. Patients and clinicians who provide these interventions may have different ideas about what encourages patients to engage in rehabilitation. It is important to understand what motivates patients and any differences in opinion between patients and clinicians. We asked patients and clinicians about the most important motivational factors. All agreed that realizing recovery is possible, setting goals or targets for the stages of recovery, and targeting interventions relevant to the patient's experience and lifestyle were the most important motivational factors. The patients also found access to medical information and being able to control the difficulty of tasks required during rehabilitation motivating. These findings could help clinicians provide rehabilitation care that is more specifically tailored to each patient's needs and preferences.

[1] Department of Physical Therapy, School of Health Sciences, Shinshu University, Nagano, Japan. [2] Department of Rehabilitation Medicine, Tokyo Bay Rehabilitation Hospital, Chiba, Japan. [3] Department of Rehabilitation Medicine, Hamamatsu University School of Medicine, Shizuoka, Japan. [4] Department of Rehabilitation, JA Nagano Kouseiren Kakeyu-Misayama Rehabilitation Center Kakeyu Hospital, Nagano, Japan. [5] Rehabilitation Center, Aichi Medical University Medical Center, Aichi, Japan. [6] Division of Occupational Therapy, Department of Rehabilitation, Nakaizu Rehabilitation Center, Shizuoka, Japan. [7] Department of Rehabilitation, Suzukake Central Hospital, Shizuoka, Japan. [8] Rehabilitation Center, Juzen Memorial Hospital, Shizuoka, Japan. [9] Department of Rehabilitation Medicine, JA Shizuoka Kohseiren Enshu Hospital, Shizuoka, Japan. [10] Department of Rehabilitation, Suzukake Healthcare Hospital, Shizuoka, Japan. [11] Department of Rehabilitation Medicine, Kakegawa Higashi Hospital, Shizuoka, Japan. [12] Department of Rehabilitation, Toyoda Eisei Hospital, Shizuoka, Japan. [13] Department of Rehabilitation, Tenryu Suzukake Hospital, Shizuoka, Japan. [14] Department of Rehabilitation Medicine, Hamakita Sakuradai Hospital, Shizuoka, Japan. [15] Department of Rehabilitation, Hamamatsu-Kita Hospital, Shizuoka, Japan. [16] Laboratory of Psychology, Hamamatsu University School of Medicine, Shizuoka, Japan. ✉email: tanakas@hama-med.ac.jp

Rehabilitation programs, including physical activity and exercise, have beneficial effects on several health outcomes for patients with physical disabilities[1]. The independent effort of the patient is necessary to sustain their rehabilitation programs, and high levels of adherence to a rehabilitation program are thought to be indicative of motivation[2,3]. In addition, a lack of motivation is often the main barrier to physical activity and exercise training[4–10]. On the basis of these reasons, clinicians working in rehabilitation are required to have knowledge of the theories and factors related to motivation[11].

According to the World Health Organization, motivation is defined as a mental function that produces the incentive to act; the conscious or unconscious driving force for action[12]. Motivation also has many other definitions[13–17]. Psychological theories of motivation suggest that motivational behavior results from a broad range of underlying factors, such as goals[18], values[19], self-determination[20], self-efficacy[21], and social relations[22]. Recently, some researchers have conceptualized motivation as emerging properties resulting from the interaction of these important factors, not as a unitary construct that can be precisely defined and assessed[23,24].

Several studies have suggested that key-motivational factors proposed in psychology also play an important role in patients' motivation for rehabilitation[4–7,9,10]. These factors include personal goals, perceived benefits of exercise, and support from family members and clinicians that can help to increase patients' adherence to rehabilitation programs. In contrast, clinical characteristics, such as health-related concerns and physical impairments, potentially decrease patients' motivation, which is a motivational problem specific to rehabilitation[4–10].

The provision of motivational strategies during rehabilitation may promote and support patient-centered care. The concept of patient-centered care is defined as care provision that is consistent with the values, needs, and desires of patients, and is achieved when clinicians involve patients in health discussions and decisions[25,26]. While patient-centered care is a core principle of evidence-based medical practice and is more likely to positively affect rehabilitation outcomes[27,28], the differences in patients' and clinicians' perceptions of motivational factors can potentially hinder patient-centered care[29]. An example of this hindrance is that, regarding goal setting, patients appear to focus on the long-term, regaining physical function and independence, and returning to former activities and roles. In contrast, clinicians' goals tend to be short-term, specific, conservative in ambition, and driven by financial and organizational pressure[29]. Additionally, with regard to rehabilitation programs, group exercise has been reported to be a perceived motivator of physical activity for individuals with stroke[4]. However, our previous Delphi study indicated that rehabilitation experts rated group rehabilitation as neither effective nor ineffective in motivating these individuals[30]. To the best of our knowledge, no studies have directly compared patients' preferences regarding motivational factors with those of clinicians. Therefore, the current study aimed to compare patients' and clinicians' perceptions of the most important factors in motivating patients for rehabilitation. We hypothesized that patients and clinicians differ to some extent in their perceptions of the relative importance of factors that motivate patients to perform rehabilitation.

As a result, we find the three motivational factors, such as the realization of recovery, goal setting, and practice related to the patient's experience and lifestyle, are endorsed by patients and clinicians. Additionally, some motivational factors are preferred by patients over clinicians. These findings may have important implications for effectively motivating patients to engage in rehabilitation.

## Methods

**Study design.** We used a multicenter explanatory survey research design. This study protocol was approved by the appropriate ethics committee at the Hamamatsu University School of Medicine (approval number: 21-233). Informed consent was obtained from all participants.

**Participants.** Patients who were hospitalized in an intensive inpatient rehabilitation ward were recruited through purposive sampling on the basis of the inclusion criteria from 12 hospitals in Japan. Intensive inpatient rehabilitation wards assist patients in acquiring skills for activities of daily living to increase the likelihood of home discharge[31,32]. All patients hospitalized in the wards meeting the inclusion criteria were referred to the research team by a researcher at each hospital. The inclusion criteria were as follows: being aged 20 to 90 years, having an established diagnosis of neurological or orthopedic disorders as the primary reason for hospitalization, having undergone an inpatient rehabilitation program for at least 4 weeks at the time of study participation, and having sufficient communication skills to complete the questionnaire with the assistance of a researcher. Demographic and clinical data, such as primary reasons for hospitalization and sex, were obtained from patient's medical records. Clinicians were purposively sampled from 13 hospitals in Japan, and included physicians, physical therapists, occupational therapists, and speech-language-hearing therapists working in the intensive inpatient rehabilitation ward. Patients were recruited as participants from 12 of these hospitals. Surveys of inpatients were not permitted in the one remaining hospital because of the coronavirus disease 2019 pandemic.

The sample size calculation for participants was based on epidemiological data from the Kaifukuki Rehabilitation Ward Association[33]. In total, 38,363 patients with neurological and orthopedic diseases were admitted to intensive inpatient rehabilitation wards. Of these patients, 18,870 had neurological diseases and 19,493 had orthopedic diseases. Similarly, the total number of rehabilitation clinicians working in the intensive inpatient rehabilitation wards was estimated to be 66,033, with 30,911 physical therapists, 18,700 occupational therapists, 8843 physicians, and 7579 speech-language-hearing therapists. Based on these population sizes and using a margin of error of 5 at a 95% confidence interval (CI), the estimated minimum sample size was 381 patients and 382 clinicians in this study[34].

**Questionnaire content.** The first author initially developed a list of potential factors involved in increasing the patients' motivation for rehabilitation on the basis of findings from our previous studies[30,35,36] and related international literature[3–6,9,37–46]. Two researchers (K.S. and S.T.) reviewed the items for clarity, relevance, and topic coverage[47]. We conducted a pilot test with a small sample of patients and clinicians to determine whether participants consistently understood the meaning of each item[48]. On the basis of feedback from the pilot test, minor grammatical changes were made. Consequently, we prepared a list of 15 potential motivational factors for rehabilitation (Supplementary Data 1). All survey data were captured anonymously for patients and clinicians.

We administered the patient questionnaire in an interview style, and patients participated in a face-to-face structured interview with a researcher at each hospital. Patients were presented with a list of 16 items, including "other" in addition to the 15 potential motivational factors. The list presented to patients did not include the specific examples shown in Supplementary Data 1 and Supplementary Table S1, because they received verbal explanations of specific examples from the

interviewer. The structured interview included two questions. In the first question, patients were asked to select the three most important factors for facilitating their engagement in rehabilitation from the list. In the second question, they were instructed to choose the most important factor from the three factors that they selected in the first question. The participants who selected "other" were asked to respond to an open-ended question in which they proposed additional motivational factors. The structured interview guide is shown in Supplementary Table S2. The interview lasted less than 5 min. The patient survey was available in paper form. At each hospital, one researcher was responsible for administering the survey. Following the end of the recruitment period, all completed questionnaires were mailed to the first author.

We used a cloud-based questionnaire and survey software (Google Forms; Google LLC, Mountain View, CA, USA) to develop the clinician survey and collect data. To publicize the study, the researcher at each hospital distributed leaflets to clinicians who met the inclusion criteria. The leaflets contained a brief description of the study and a hyperlink to the survey. The clinicians could voluntarily access the survey website using their own laptops, tablets, or smartphones. The survey included two questions and several demographic characteristics. The survey questions that were provided to clinicians are shown in Supplementary Data 2. In the first question, clinicians were asked to select the three most important factors for increasing the patients' adherence to rehabilitation programs from the list shown in Supplementary Data 1. In the second question, the clinicians were instructed to choose the most important factor of the three that they selected in the first question. The clinicians who completed the survey were reimbursed for their participation with a 500 JPY gift card (approximately 3.50 USD).

**Statistics and reproducibility**. The primary outcome of the study was the patients' and clinicians' top choice for the most important motivational factor. The secondary outcome included the three factors that they selected in the first question. We used descriptive statistics to summarize the demographic characteristics of the patients and clinicians and their responses to the two survey questions. We used a margin of error of 5% to determine the sample size in this study. Therefore, motivational factors selected by more than 5% of participants in each of the patient and clinician groups were considered to constitute their preferences. We compared patients' responses with those of clinicians using Fisher's exact test. Phi coefficients with 95% CIs were calculated as the measure of effect size for comparing responses between groups with the following equation:

$$\text{Phi coefficient} = (A \times D - B \times C) / \{(A + B) \times (C + D) \times (A + C) \times (B + D)\}^{1/2}$$

where A is the number of clinicians who selected a motivational factor, B is the number of clinicians who did not select this factor, C is the number of patients who selected this factor, and D is the number of patients who did not select this factor[49]. Negative values of the phi coefficient indicate that a higher proportion of patients than clinicians rated the factor as important/most important. An absolute value of a phi coefficient of 0.05 was considered as a weak effect size, 0.10 a moderate effect size, 0.15 a strong effect size, and 0.25 a very strong effect size[50]. In addition, a multiple logistic regression analysis was used to examine the association between patients' choices regarding the most important motivational factor with their demographic characteristics, such as the primary reason for hospitalization, sex, age ≥65 years or not[51], and the length of hospital stay. Statistical analyses were performed using the Statistical Package for the Social Sciences software version 27.0 (International Business Machines Corp.,

Armonk, NY, USA). A power analysis showed that a minimum of 381 patients and 382 clinicians were required. Therefore, we considered that a similar number of participants should be included for reproducibility.

**Reporting summary**. Further information on research design is available in the Nature Portfolio Reporting Summary linked to this article.

## Results

**Participants' characteristics**. The survey was conducted from January to March 2022. Of the 520 patients who met the inclusion criteria, 23 refused to participate in this study. Consequently, we obtained data from 479 patients. In addition, of the 645 clinicians who met the inclusion criteria, 401 responded. Therefore, the response rates of the patient and clinician surveys were 92.1 and 62.2%, respectively. The demographic characteristics of the patients and clinicians are presented in Supplementary Data 3. The primary reason for hospitalization of most patients was either stroke (45.5%) or fracture (42.2%). Approximately half (49.9%) of the clinicians were physical therapists (49.9%).

**Comparison of patients' and clinicians' perceptions of the most important factors for motivating patients to engage in rehabilitation**. The distribution of patients' and clinicians' answers, when asked to report their top choice regarding the most important motivational factor, is shown in Fig. 1. The three most frequently selected motivational factors were identical for patients and clinicians: realization of recovery was chosen by 26.5% of patients and 36.7% of clinicians, goal setting was chosen by 15.0% of patients and 22.4% of clinicians, and practice related to the patient's experience and lifestyle was chosen by 10.4% of patients and 9.5% of clinicians.

Although nine motivational factors were selected by more than 5% of the patients, only five were chosen by more than 5% of the clinicians (Fig. 1). This finding indicated that patients exhibited more varied preferences for motivational factors than clinicians. The phi coefficients for comparing patients' top choices with those of clinicians are shown in Fig. 2. Of the nine motivational factors selected by more than 5% of patients, medical information ($p < 0.001$; phi $= -0.14$; 95% CI $= -0.20$ to $-0.07$) and control of task difficulty ($p = 0.011$; phi $= -0.09$; 95% CI $= -0.16$ to $-0.02$) were chosen by a significantly higher proportion of patients than clinicians. In contrast, a significantly higher proportion of clinicians than patients rated realization of recovery ($p = 0.001$; phi $= 0.11$; 95% CI $= 0.04$ to $0.18$) and goal setting ($p = 0.005$; phi $= 0.10$; 95% CI $= 0.03$ to $0.16$) as the most important.

**Comparison of patients' and clinicians' perceptions of the three most important motivational factors**. The patients' and clinicians' answers for their choice of the three most important factors among 15 potential motivational factors are shown in Fig. 3. Twelve (2.5%) patients selected "other" as one of the three most important motivational factors. Additional motivational factors proposed by the patients are shown in Supplementary Table S3. Similar to the results regarding the motivational factors perceived as the most important, the three most frequently endorsed motivational factors were identical for patients and clinicians: realization of recovery was chosen by 47.4% of patients and 59.4% of clinicians, goal setting was chosen by 34.7% of patients and 52.6% of clinicians, and practice related to the patient's experience and lifestyle was chosen by 32.4% of patients and 38.2% of clinicians.

The phi coefficients for comparing patients' choices with those of clinicians for the first question of the survey are shown in

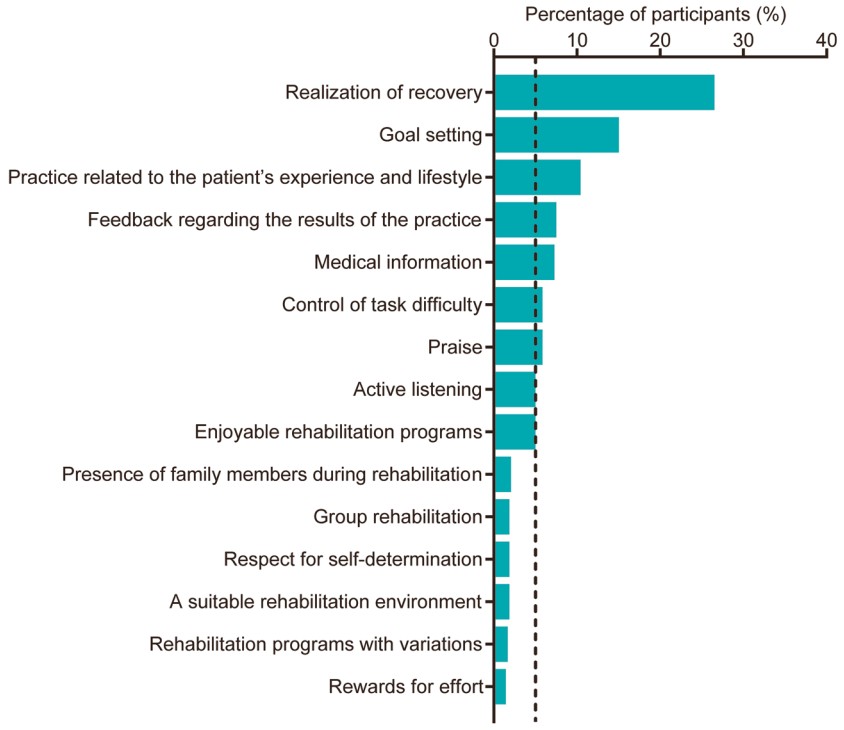

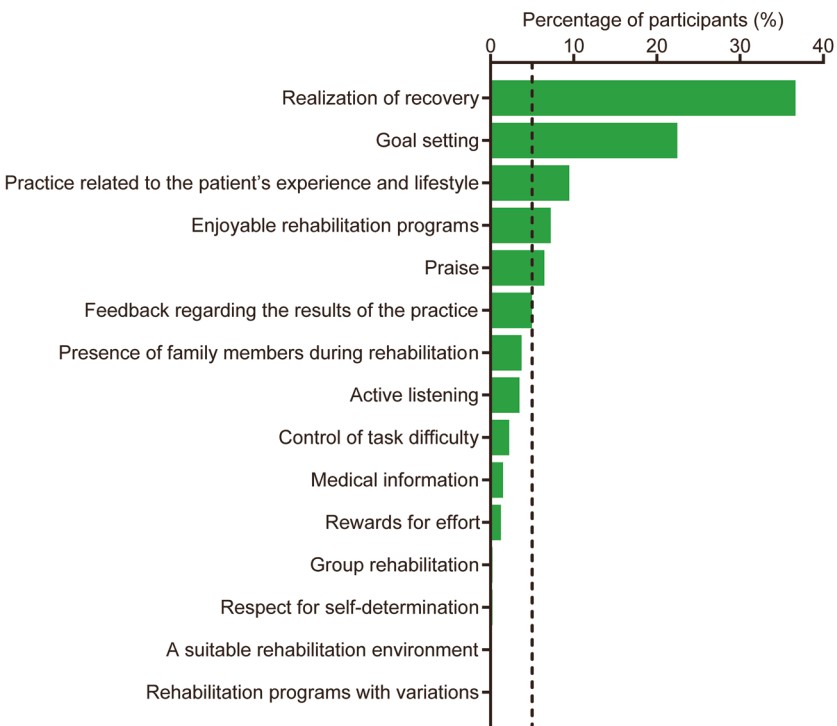

**Fig. 1 Distribution of participants' answers regarding the most important motivational factor. a** Distribution of patients' answers (*n* = 476).
**b** Distribution of clinicians' answers (*n* = 401). The potential motivational factors are arranged in descending order by the percentage of participants who selected each factor as the most important. The vertical dashed line represents 5% of participants.

Fig. 4. A significantly higher proportion of patients than clinicians rated a suitable rehabilitation environment ($p < 0.001$; phi = $-0.19$; 95% CI = $-0.26$ to $-0.12$), rehabilitation programs with variations ($p < 0.001$; phi = $-0.16$; 95% CI = $-0.23$ to $-0.09$), control of task difficulty ($p < 0.001$; phi = $-0.15$; 95% CI: $-0.22$ to $-0.09$), respect for self-determination ($p < 0.001$; phi = $-0.14$; 95% CI = $-0.21$ to $-0.07$), and medical information ($p < 0.001$; phi = $-0.14$; 95% CI = $-0.20$ to $-0.07$) as important. In

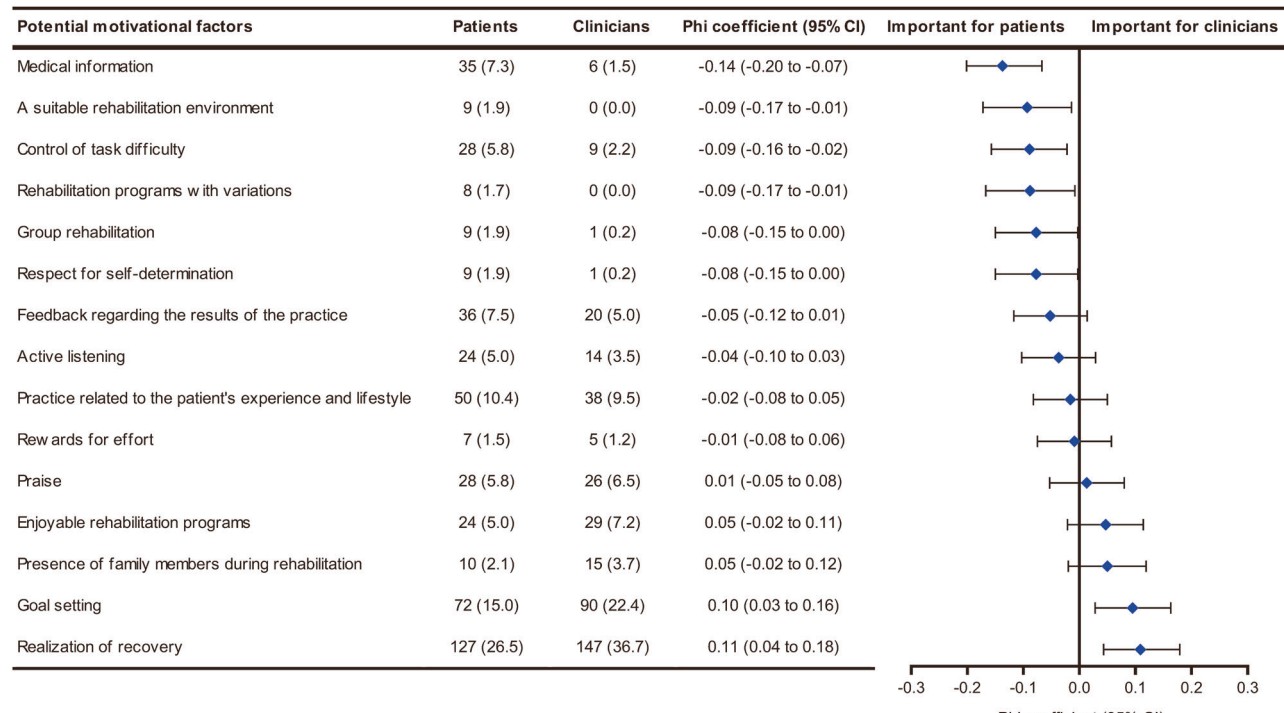

**Fig. 2 Comparison of patients' and clinicians' perceptions of the most important motivational factors.** The total number of patient and clinician participants were 476 and 401, respectively. The potential motivational factors are arranged in ascending order regarding the value of the Phi coefficient. Filled diamonds and error bars represent phi coefficients and 95% confidence intervals, respectively. Values are the number (%) unless indicated. CI confidence interval.

contrast, the presence of family members during rehabilitation ($p = 0.002$; phi $= 0.19$; 95% CI $= 0.12$ to 0.26), goal setting ($p < 0.001$; phi $= 0.18$; 95% CI: 0.12 to 0.25), enjoyable rehabilitation programs ($p < 0.001$; phi $= 0.13$; 95% CI $= 0.06$ to 0.19), and realization of recovery ($p < 0.001$; phi $= 0.12$; 95% CI $= 0.05$ to 0.19) were chosen by a significantly higher proportion of clinicians than patients.

**Associations between patients' choices regarding the most important motivational factors and their demographic characteristics.** In patients with stroke and those with fracture ($n = 420$), who comprised the majority of the participants (87.7%), we evaluated the associations between the patient's choices regarding the most important motivational factor and their demographic characteristics. Most patients were those with stroke ($n = 218$; 51.9%), female ($n = 240$; 57.1%), and those aged ≥65 years ($n = 352$; 84.3%). The median length of hospital stay was 43 days (interquartile range, 34 to 64 days). The results of the multiple logistic regression analysis are shown in Supplementary Data 4. A significantly higher proportion of patients aged <65 years old ($n = 24$; 36.2%) rated realization of recovery ($p = 0.039$; odds ratio $= 0.53$; 95% CI $= 0.29$ to 0.97) as the most important factor than those aged ≥65 years old ($n = 84$; 23.8%). In addition, goal setting ($p = 0.031$; odds ratio $= 0.46$; 95% CI $= 0.22$ to 0.93) was also chosen by a significantly larger proportion of patients aged <65 years old ($n = 15$; 22.7%) than those aged ≥65 years old ($n = 50$; 8.5%). Furthermore, patients with a shorter length of hospital stay were significantly more likely to choose medical information as the most important factor ($p = 0.027$; odds ratio $= 0.97$; 95% CI $= 0.94$ to 1.00), although only 27 of 420 patients chose this factor.

**Discussion**
To the best of our knowledge, the present study is the first to investigate the similarities and differences between patients' and

clinicians' perceptions of the most important factors for motivating patients to engage in rehabilitation. The three motivational factors most frequently endorsed by patients were identical to those endorsed by clinicians. Furthermore, patients had more diverse preferences for motivational factors than clinicians. These findings broaden our understanding of motivation in patient-centered rehabilitation. Additionally, because there are no guidelines and adequate training programs for clinicians regarding motivational strategies[35,52], our findings may provide clinicians with helpful information for effectively motivating patients to engage in rehabilitation.

One of the main findings in this study regarding similarity in perceptions was that not only clinicians, but also patients, considered goal setting and practice related to the patient's experience and lifestyle as the most important factors. The results of our supplemental analysis, obtained by repeated random sampling with replacement from all participants, also support the reliability of this finding (Supplementary Figs. S1, S2). Previous theoretical, experimental, and observational studies have reported that these two factors are essential components of motivation[11,30,35,36,53–55]. In addition, goal setting is regarded as one of the important procedures to facilitate an interdisciplinary team approach[31]. Therefore, many clinicians use these factors as key-motivational strategies in intensive inpatient rehabilitation wards. During rehabilitation, patients may share the intentions of clinicians and recognize them as important for overcoming difficulties. Our study suggests that these strategies can be core motivators, supported not only by medical evidence, but also by subjective perceptions of patients and clinicians. Another main finding in this study is that realization of recovery was considered to be the most important factor by patients and clinicians. The realization of recovery is associated with positive achievement emotions and/or self-efficacy that follow success in rehabilitation[21,56]. Several studies have indicated that patients' emotions at the initial stage

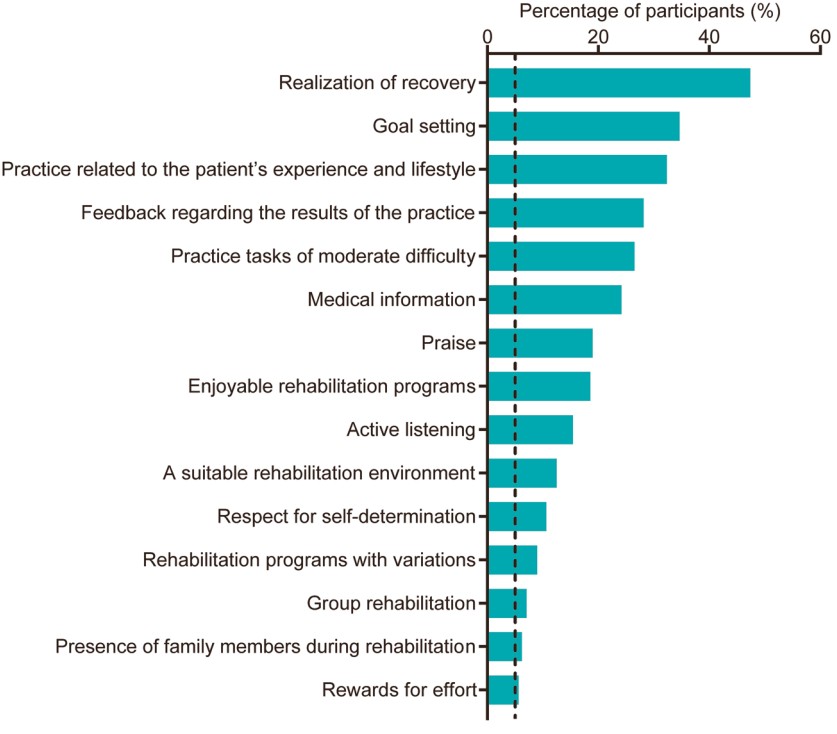

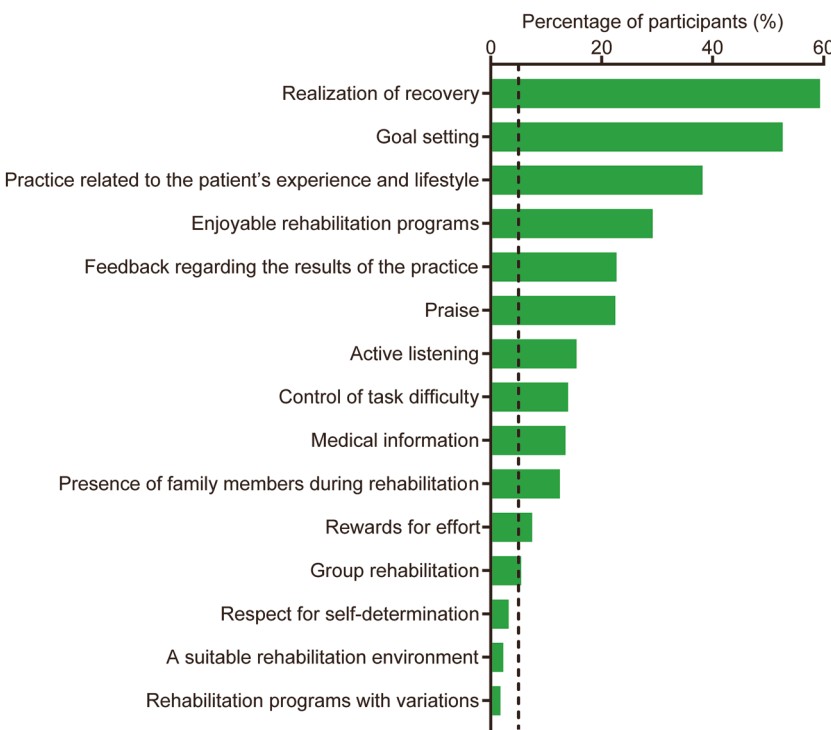

**Fig. 3 Distribution of participants' answers regarding the three most important motivational factors. a** Distribution of patients' answers (*n* = 479). **b** Distribution of clinicians' answers (*n* = 401). The potential motivational factors are arranged in descending order by the percentage of participants who selected each factor as the most important. The vertical dashed line represents 5% of participants.

of rehabilitation predict their subsequent motor performance and recovery after brain injury[57–59]. Studies have also shown that successful rehabilitation influences the development of positive emotions and self-efficacy in a real rehabilitation setting[60–62].

Therefore, the relationship between the achievement of emotions/self-efficacy and outcomes would be reciprocal rather than unidirectional in rehabilitation. Our results indicate that patients and clinicians may recognize the importance of the realization of

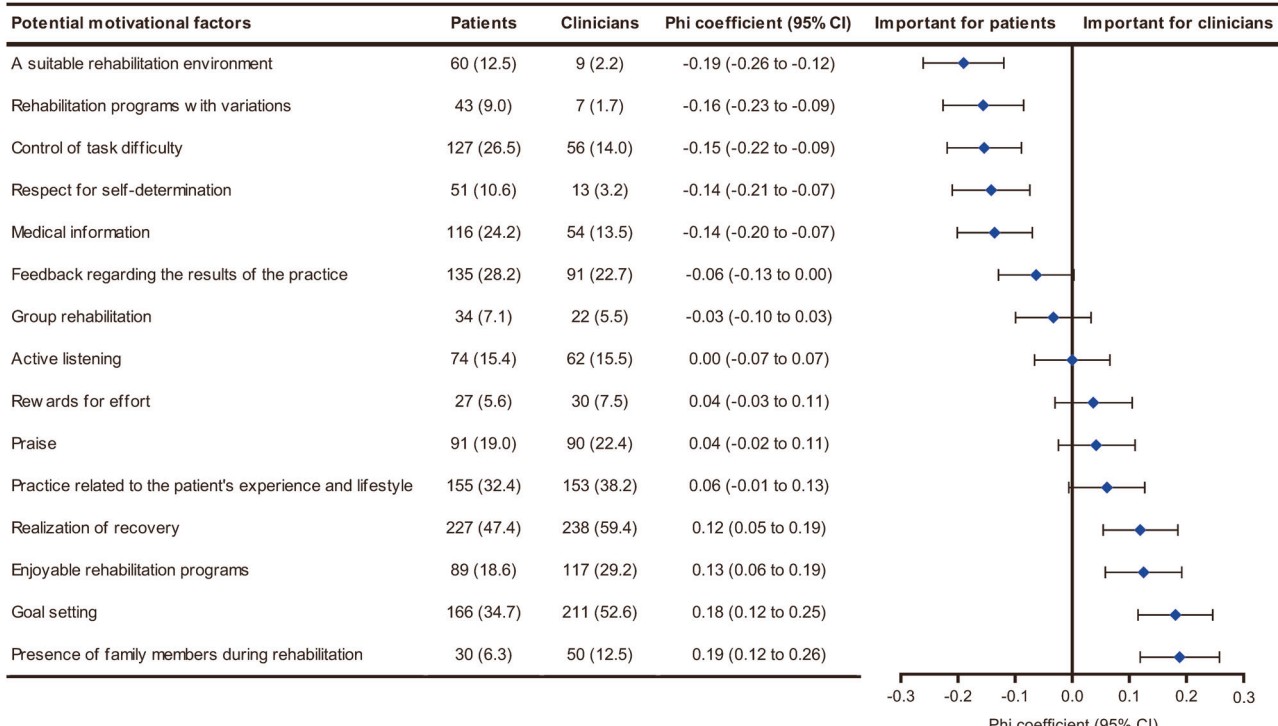

| Potential motivational factors | Patients | Clinicians | Phi coefficient (95% CI) | Important for patients | Important for clinicians |
|---|---|---|---|---|---|
| A suitable rehabilitation environment | 60 (12.5) | 9 (2.2) | -0.19 (-0.26 to -0.12) | | |
| Rehabilitation programs with variations | 43 (9.0) | 7 (1.7) | -0.16 (-0.23 to -0.09) | | |
| Control of task difficulty | 127 (26.5) | 56 (14.0) | -0.15 (-0.22 to -0.09) | | |
| Respect for self-determination | 51 (10.6) | 13 (3.2) | -0.14 (-0.21 to -0.07) | | |
| Medical information | 116 (24.2) | 54 (13.5) | -0.14 (-0.20 to -0.07) | | |
| Feedback regarding the results of the practice | 135 (28.2) | 91 (22.7) | -0.06 (-0.13 to 0.00) | | |
| Group rehabilitation | 34 (7.1) | 22 (5.5) | -0.03 (-0.10 to 0.03) | | |
| Active listening | 74 (15.4) | 62 (15.5) | 0.00 (-0.07 to 0.07) | | |
| Rewards for effort | 27 (5.6) | 30 (7.5) | 0.04 (-0.03 to 0.11) | | |
| Praise | 91 (19.0) | 90 (22.4) | 0.04 (-0.02 to 0.11) | | |
| Practice related to the patient's experience and lifestyle | 155 (32.4) | 153 (38.2) | 0.06 (-0.01 to 0.13) | | |
| Realization of recovery | 227 (47.4) | 238 (59.4) | 0.12 (0.05 to 0.19) | | |
| Enjoyable rehabilitation programs | 89 (18.6) | 117 (29.2) | 0.13 (0.06 to 0.19) | | |
| Goal setting | 166 (34.7) | 211 (52.6) | 0.18 (0.12 to 0.25) | | |
| Presence of family members during rehabilitation | 30 (6.3) | 50 (12.5) | 0.19 (0.12 to 0.26) | | |

**Fig. 4 Comparison of patients' and clinicians' perceptions of the three most important motivational factors.** The total number of patient and clinician participants were 479 and 401, respectively. The potential motivational factors are arranged in ascending order regarding the value of the phi coefficient. Filled diamonds and error bars represent phi coefficients and 95% confidence intervals, respectively. Values are the number (%) unless indicated. CI confidence interval.

recovery to develop the positive reciprocal process in rehabilitation.

Of nine motivational factors endorsed by more than 5% of patients as the most important, medical information and control of task difficulty were preferred more by patients than by clinicians. Previous qualitative studies of individuals with physical disabilities reported that information regarding the benefits of rehabilitation programs is an important factor in increasing patients' motivation[6,9,36,37,42,46,63]. Additionally, rehabilitation programs that combine exercise therapy with information provision and therapeutic patient education of patients have shown positive outcomes in patients with a range of neurological and orthopedic disorders[64–71]. Regarding the influence of the level of task difficulty on motivation, an experimental study reported that participants exert less effort in difficult trials compared with easy trials[72]. Failure of feedback has been reported to undermine learning motivation because it decreases people's confidence in their overall ability to pursue their goals and their general expectations of success[73]. Therefore, strategies, such as explaining the rehabilitation process and providing a practice task that is achievable with little effort, may be more effective in motivating patients than clinicians believe. Consequently, when determining which motivational strategies to use, clinicians should consider individual patient's preferences regarding motivational factors. Patients' information, such as demographic characteristics and personality attributes, and a patient's reactions to a presented motivational strategy, may help clinicians better understand patient's preferences.

This study suggests that preferences for motivational factors vary depending on the patient's age and length of hospital stay. We found that goal setting and realization of recovery were preferred by relatively younger patients (<65 years of age) than older patients. Goal setting has been shown to be a more important motivator for physical activity in younger people than in older people[74]. In our previous qualitative study of physical therapists, participants listed setting goals, such as returning to work and society, as an effective motivational strategy for relatively young patients[36]. In addition, younger people with physical disabilities tend to have higher expectations of what they can achieve, such as wanting to be able to participate in sporting activities or to lead an active social life, than other age groups[75]. Furthermore, a qualitative study with patients with stroke in the intensive inpatient rehabilitation ward suggested that improvement in physical function had a more positive effect on motivation in relatively younger patients than in older patients[60]. These previous findings support the results of the present study. Therefore, setting goals that are useful to patients and helping them experience positive achievement emotions may be especially important for enhancing active participation in rehabilitation for relatively young patients.

Additionally, patients with shorter hospital stays were more likely to consider medical information to be the most important motivational factor. This result suggests that interventions, such as information provision[64–66] and therapeutic patient education[67–71], are effective for increasing the motivation of patients in the early period after admission to the intensive inpatient rehabilitation ward. These findings may help clinicians use different motivational strategies tailored to the patients' conditions.

The limited combinations of patients' choices and their demographic characteristics that showed statistically significant associations may be explained by the small sample size. The number of patients who selected the 12 motivational factors that were not significantly associated with any of the demographic variables was less than 50, which resulted in large CIs for the odds ratio. Therefore, because the small sample size could have reduced the power to detect statistically significant associations

between patients' choices and their demographic characteristics, careful interpretation of the results of multiple logistic regression analyses is necessary.

A primary limitation of this study is that all of the participants were recruited in Japan, potentially limiting the international generalizability of our findings. Nevertheless, the present results support the results of previous studies conducted in different countries[3–6,9,37–46]. An international survey would improve the external validity of our findings. Another potential limitation is that the opinions of patients with stroke and patients with fractures might have been overstated in the current sample. Similarly, the responses of physical therapists may have been overstated in clinicians' perceptions. However, patients with stroke and patients with fracture account for approximately 80% of inpatients in intensive inpatient rehabilitation wards in Japan[33]. Additionally, among rehabilitation clinicians working in these wards, 45.7% are physical therapists, 28.6% are occupational therapists, 14.3% are physicians, and 11.4% are speech-language-hearing therapists[33]. Therefore, the percentage of patients with stroke, patients with fractures, and physical therapists in our sample is likely to be consistent with the actual situation.

In conclusion, the three motivational factors most frequently selected as the most important by patients were identical to those selected by clinicians, suggesting the existence of core motivational strategies that are considered to be important by patients and clinicians. Additionally, patients exhibited more diverse preferences regarding motivational factors than clinicians, revealing some motivational factors that were preferred by patients over clinicians. Therefore, clinicians are required to consider individual patients' preferences to promote patient-centered care in rehabilitation when determining the most appropriate strategies.

### Data availability

The datasets used and/or analyzed during the current study cannot be made publicly available due to the need for participant confidentiality. However, they are available from the corresponding author on reasonable request. The numerical data underlying Figs. 1 and 3 are in Supplementary Data 5 and 6, respectively.

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

## Acknowledgements

The authors thank Masafumi Sugasawa, Tsuyoshi Tatemoto, Yuto Goto, and Ayumi Mochida at the Tokyo Bay Rehabilitation Hospital for their help and support. This work was supported by a JSPS KAKENHI grant (JP20K21752) and HUSM Grant-in-Aid to Satoshi Tanaka. The funder had no role in the design and conduct of the study; collection, management, analysis, and interpretation of the data; preparation, review, or approval of the manuscript; and decision to submit the manuscript for publication.

## Author contributions

K.O. and S.T. contributed to conceptualization, methodology, validation, data curation, formal analysis, writing, and visualization. K.Y., S.I., K.S., H.O., J.I., T.E., T.O., M.H., Y.H., A.H., K.Y., Y.K., K.S., A.A., and T.N. contributed to investigation and resources. K.O., K.Y., and S.T. contributed to supervision. S.T. contributed to project administration and funding acquisition. All authors gave final approval of the version to be published.

## Competing interests

The authors declare no competing interests.
