## [Peer Review File · Communications Medicine]

Reviewers' comments:

Reviewer #1 (Remarks to the Author):

The submitted manuscript is about motivational factors in rehabilitation that are investigated in patients and clinicians in a cross-sectional survey study. The topic of motivation is an important factor for successful and efficient rehabilitation programs and the study is generally well thought out and with clearly presented methods and results that are relevant to researchers and clinicians alike. However the background on motivation seems to be a bit too sketchy.

The following points should be addressed in a revision before acceptance can be recommended:

- lines 43-48: The introduction and definition of motivation is quite short. Could the authors provide more information on factors contributing to motivation and associated concepts, such as self-efficacy to provide a more complete picture to the readers?
- line 67: What is the institution name?
- line 71: consider rewording of "convalescent rehabilitation ward" were participants of inpatient and / or outpatient rehabilitation programs included?
- line 75: consider rewording of "all rehabilitation inpatients"
- line 80: what is specifically meant by "adequate communication skills to complete the questionnaire"? Independent completion or was reading assistance allowed?
- line 85: Why did the authors recruited patients from 12 hospitals and not 13 as the clinicians?
- lines 170-171, 189-190: what do the ranges of percentages refer to?

Reviewer #2 (Remarks to the Author):

Thank you for your submission on an important topic in rehabilitation. Significant revisions are recommended.

Introduction - this section needs significant attention. The first 3 sentences of manuscript talk about physical activity where this is a manuscript that describes perceptions of motivations and offers and comparison between clinicians and therapists. This should be clear when reading the introduction and build a case for why this work is necessary.

Methodology - I wouldn't describe this as a cross sectional study instead consider this survey research. There are aspects that are explanatory survey research.

Data analysis - The odds ratio doesn't fit well for me. Because I don't see the distinction between patient and therapist as 'exposure' I don't see the fit for the odds ratio in the primary analysis. I would like a strong justification for the choice of this method. The introduction hypothesized that there was a difference between clinicians and patients - is the odds ratio the best way to show there is a difference? The results section state the perceptions are similar but don't provide the statistics to fully support those statements. I see the value of odds ratio related to patient characteristics and your regression analysis.

Results - some of the content is better placed in discussion. For example line 239 citation of other work would be better in a discussion where you can compare and contrast your results with others. As the reader I would benefit from a stronger focus on the statistics you obtained here. I want the tables to be fully explained

Discussion

224-231 do not belong in discussion section

283-288- need to either better support or cite statements like this.

Tables

Is Table 1 the actual survey that was given? Did the participants have these examples?

Provide the actual survey questions that were read to patients the written survey given to the clinicians

Tabel 3 - there were very limited sig associations. Why do you think this is? Is this a problem in the model?

Specific question

Line 171 I'm not clear what those ranges mean. I would like to see the full data in a table

Thank you for the opportunity to review.

Response to the reviewers' comments

To Reviewer #1

We thank the reviewer for the time they devoted to reviewing our manuscript, and for their helpful comments. We have addressed each comment in a point-by-point manner below. In addition, we have used a native English proofreading service again to refine our manuscript and have made some corrections in the manuscript according to the proofreaders' suggestions. Please see the revised manuscript for details.

● Comment 1

Lines 43-48: The introduction and definition of motivation is quite short. Could the authors provide more information on factors contributing to motivation and associated concepts, such as self-efficacy to provide a more complete picture to the readers?

Response

We appreciate the reviewer's helpful comment and have added the following sentences in the Introduction section to provide more information regarding the definition of motivation and the factors contributing to it, including self-efficacy.

Page 3, line 34 to page 4, line 55

“Rehabilitation programs including physical activity and exercise have beneficial effects on several health outcomes for patients with physical disabilities¹. The independent effort of the patient is necessary to sustain their rehabilitation programs, and high levels of adherence to a rehabilitation program are thought to be indicative of motivation^{2,3}. In addition, a lack of motivation is often the main barrier to physical activity and exercise training⁴⁻¹⁰. For these reasons, clinicians working in rehabilitation are required to have knowledge of the theories and factors related to motivation¹¹.

According to the World Health Organization, motivation is defined as “mental function that produces the incentive to act; the conscious or unconscious driving force for action¹².” Motivation also has many other definitions¹³⁻¹⁷. Psychological theories of motivation suggest that motivational behavior results from a broad range of underlying factors such as goals¹⁸, values¹⁹, self-determination²⁰, self-efficacy²¹, and social relations²². Recently, some researchers have conceptualized motivation as emerging properties resulting from the interaction of these key factors, not as a unitary construct that can be precisely defined and assessed^{23,24}.

Several studies suggest that key motivational factors proposed in psychology also play an important role in patients' motivation for rehabilitation^{4-7,9,10}. For example, personal goals, perceived benefits of exercise, and support from family members and clinicians can help to increase patients' adherence to rehabilitation programs. In contrast, clinical characteristics such as health-related concerns and physical impairments potentially decrease patient motivation, which is a motivational problem specific to rehabilitation⁴⁻¹⁰.”

In addition, to discuss the relationship between self-efficacy that you noted and the results of this study, we have added the following sentences to the Discussion section.

Page 16, lines 271–280

““Realization of recovery” is associated with positive achievement emotions

and/or self-efficacy that follow success in rehabilitation^{21,43}. Several studies indicate that patients' emotions at the initial stage of rehabilitation predict their subsequent motor performance and recovery after brain injury⁴⁴⁻⁴⁶. Studies also indicate that successful rehabilitation influences the development of positive emotions and self-efficacy in a real rehabilitation setting⁴⁷⁻⁴⁹. Therefore, the relationship between achievement emotions/self-efficacy and outcomes would be reciprocal rather than unidirectional in rehabilitation. Our results indicate that both patients and clinicians may recognize the importance of "realization of recovery" in order to develop the positive reciprocal process in rehabilitation."

Furthermore, we have added the following references.

- 2 Maclean, N., Pound, P., Wolfe, C. & Rudd, A. Qualitative analysis of stroke patients' motivation for rehabilitation. *BMJ* **321**, 1051-1054 (2000).
- 3 Maclean, N., Pound, P., Wolfe, C. & Rudd, A. The concept of patient motivation: A qualitative analysis of stroke professionals' attitudes. *Stroke* **33**, 444-448 (2002).
- 14 Kim, S. I. Neuroscientific model of motivational process. *Front. Psychol.* **4**, 98; 10.3389/fpsyg.2013.00098 (2013).
- 15 Pessiglione, M., Vinckier, F., Bouret, S., Daunizeau, J. & Le Bouc, R. Why not try harder? Computational approach to motivation deficits in neuro-psychiatric diseases. *Brain* **141**, 629-650 (2018).
- 16 Simpson, E. H. & Balsam, P. D. The behavioral neuroscience of motivation: An overview of concepts, measures, and translational applications. *Curr. Top. Behav. Neurosci.* **27**, 1-12 (2016).
- 17 Vu, T. *et al.* Motivation-achievement cycles in learning: A literature review and research agenda. *Educ. Psychol. Rev.* **34**, 39-71 (2022).
- 18 Ryan, T. A. *Intentional behavior: An approach to human motivation.* (Ronald Press, 1970).
- 19 Raynor, J. O. & McFarlin, D. B. Foundations of social behavior in *Handbook of motivation and cognition* (eds. Sorrentino, R.M. & Higgins, E.T.) 315-349 (Guilford Press, 1986).
- 20 Deci, E. L. & Ryan, R. M. Self-determination theory in health care and its relations to motivational interviewing: A few comments. *Int. J. Behav. Nutr. Phys. Act.* **9**, 24; 10.1186/1479-5868-9-24 (2012).
- 21 Schunk, D. H. Self-efficacy and academic motivation. *Educ. Psychol.* **26**, 207-231 (1991).
- 22 Wentzel, K. R. Social relationships and motivation in middle school: The role of parents, teachers, and peers. *J. Educ. Psychol.* **90**, 202-209 (1998).
- 23 Tamura, A. *et al.* Exploring the within-person contemporaneous network of motivational engagement. *Learn. Instr.* **81**, 101649; 10.1016/j.learninstruc.2022.101649 (2022).
- 24 Murayama, K. & Elliot, A. J. The competition-performance relation: A meta-analytic review and test of the opposing processes model of competition and performance. *Psychol. Bull.* **138**, 1035-1070 (2012).
- 48 McGrane, N., Galvin, R., Cusack, T. & Stokes, E. Addition of motivational interventions to exercise and traditional physiotherapy: a review and meta-analysis. *Physiotherapy* **101**, 1-12 (2015).
- 49 Cheng, D. *et al.* Motivational interviewing for improving recovery after stroke. *Cochrane Database Syst Rev*, CD011398 (2015).

- **Comment 2**
Line 67: What is the institution name?

Response

“*Institution Name*” refers to the name of the institution at which ethics committee approval was obtained. Because we have requested a Double Anonymous Peer Review process, we used the term “*Institution Name*” in the manuscript to conceal the authors’ identities from reviewers.

- **Comment 3**
Line 71: consider rewording of "convalescent rehabilitation ward" were participants of inpatient and/or outpatient rehabilitation programs included?

Response

According to the reviewer’s suggestion, we have used the term “**intensive inpatient rehabilitation ward**” instead of “convalescent rehabilitation ward” throughout the manuscript. Please see the revised manuscript for the change. Therefore, all patients who participated in this study were offered inpatient rehabilitation programs, not outpatient rehabilitation programs. In addition, we have added the following sentence to explain the role of the intensive inpatient rehabilitation ward.

Page 5, lines 83–84

“**Intensive inpatient rehabilitation wards assist patients in acquiring skills for activities of daily living to increase the likelihood of home discharge^{31,32}.**”

- **Comment 4**
Line 75: consider rewording of "all rehabilitation inpatients."

Response

Thank you for pointing it out. We have reworded “All rehabilitation inpatients” to “**All patients hospitalized in the wards**”. Please see the revised sentence (page 5, lines 84–85).

- **Comment 5**
Line 80: what is specifically meant by "adequate communication skills to complete the questionnaire"? Independent completion or was reading assistance allowed?

Response

The phrase “adequate communication skills to complete the questionnaire” refers to the ability to complete the questionnaire with a researcher’s assistance. We have rewritten this sentence to clarify our meaning as follows.

Page 6, lines 89–91

“Inclusion criteria were as follows: being aged 20 to 90 years, having an established diagnosis of neurological or orthopedic disorders as the primary reason for hospitalization, having undergone an inpatient rehabilitation program

for at least 4 weeks at the time of study participation, and having sufficient communication skills to complete the questionnaire with the assistance of a researcher.”

● **Comment 6**

Line 85: Why did the authors recruited patients from 12 hospitals and not 13 as the clinicians?

Response

In one of the 13 hospitals where clinicians were recruited, surveys of inpatients were not permitted because of restrictions related to the coronavirus disease 2019 pandemic. Hence, patients were recruited from 12 of the 13 hospitals. We have added the following sentence in the Methods section to clarify this point.

Page 6, lines 96–97

“Patients were recruited as participants from 12 of these hospitals. Surveys of inpatients were not permitted in the one remaining hospital because of the coronavirus disease 2019 pandemic.”

● **Comment 7**

Lines 170-171, 189-190: what do the ranges of percentages refer to?

Response

The ranges of percentages refer to the proportions of patients and clinicians who chose “realization of recovery,” “goal setting,” and “practice related to the patient’s experience and lifestyle.” We have added the percentages of participants who chose these factors in the Results section, as follows.

Page 11, lines 183–188

“The three most frequently selected motivational factors were identical for patients and clinicians: “realization of recovery” chosen by 26.5% of patients and 36.7% of clinicians, “goal setting” chosen by 15.0% of patients and 22.4% of clinicians, and “practice related to the patient’s experience and lifestyle” chosen by 10.4% of patients and 9.5% of clinicians.”

Page 12, lines 209–212

“Similar to the results regarding the motivational factors perceived as the most important, the three most frequently endorsed motivational factors were identical for patients and clinicians: “realization of recovery” chosen by 47.4% of patients and 59.4% of clinicians, “goal setting” chosen by 34.7% of patients and 52.6% of clinicians, and “practice related to the patient’s experience and lifestyle” chosen by 32.4% of patients and 38.2% of clinicians.”

In addition, according to Reviewer #2’s comment 9, we have added the percentages of participants who selected the relevant item to Figures 2 and 4. Please see the revised Figures 2 and 4.

We have also changed the legends of Figures 2 and 4, as follows.

Page 34, lines 581–583

“Figure 2. Comparison of patients’ and clinicians’ perceptions of the most important motivational factors. Motivational factors are arranged in ascending order regarding the value of the **phi coefficient**. Filled diamonds and **error bars** represent **phi coefficients** and 95% confidence intervals, respectively. **Values are number (%) unless indicated.**”

Page 35, lines 591–594

“Figure 4. Comparison of patients’ and clinicians’ perceptions of the three most important motivational factors. Motivational factors are arranged in ascending order regarding the value of the **phi coefficient**. Filled diamonds and error bars represent **phi coefficients** and 95% confidence intervals, respectively. **Values are number (%) unless indicated.**”

To Reviewer #2

Thank you for your time and helpful comments. We have addressed each comment in a point-by-point manner below. Additionally, we have used a native English proofreading service again to refine our manuscript and have revised some parts of the manuscript following the proofreaders' suggestions. Please see the revised manuscript for details.

- **Comment 1**

Introduction - this section needs significant attention. The first 3 sentences of manuscript talk about physical activity where this is a manuscript that describes perceptions of motivations and offers and comparison between clinicians and therapists. This should be clear when reading the introduction and build a case for why this work is necessary.

Response

Thank you for your helpful comment. We have re-configured the overall structure of the Introduction section to emphasize the rationale of the study according to your suggestions, as follows.

Page 3, line 34 to page 5, line 69

“Rehabilitation programs including physical activity and exercise have beneficial effects on several health outcomes for patients with physical disabilities¹. The independent effort of the patient is necessary to sustain their rehabilitation programs, and high levels of adherence to a rehabilitation program are thought to be indicative of motivation^{2,3}. In addition, a lack of motivation is often the main barrier to physical activity and exercise training⁴⁻¹⁰. For these reasons, clinicians working in rehabilitation are required to have knowledge of the theories and factors related to motivation¹¹.

According to the World Health Organization, motivation is defined as “mental function that produces the incentive to act; the conscious or unconscious driving force for action¹².” Motivation also has many other definitions¹³⁻¹⁷. Psychological theories of motivation suggest that motivational behavior results from a broad range of underlying factors such as goals¹⁸, values¹⁹, self-determination²⁰, self-efficacy²¹, and social relations²². Recently, some researchers have conceptualized motivation as emerging properties resulting from the interaction of these key factors, not as a unitary construct that can be precisely defined and assessed^{23,24}.

Several studies suggest that key motivational factors proposed in psychology also play an important role in patients' motivation for rehabilitation^{4-7,9,10}. For example, personal goals, perceived benefits of exercise, and support from family members and clinicians can help to increase patients' adherence to rehabilitation programs. In contrast, clinical characteristics such as health-related concerns and physical impairments potentially decrease patient motivation, which is a motivational problem specific to rehabilitation⁴⁻¹⁰.

The provision of motivational strategies during rehabilitation may promote and support patient-centered care. The concept of patient-centered care is defined as “care provision that is consistent with the values, needs, and desires of patients, and is achieved when clinicians involve patients in health discussions and decisions^{25,26}.” Although patient-centered care is a core principle of evidence-based medical practice and is more likely to positively affect rehabilitation outcomes^{27,28}, the differences in patients' and clinicians' perceptions of

motivational factors can potentially hinder patient-centered care²⁹. For example, although group exercise has been reported to be a perceived motivator of physical activity for individuals with stroke⁴, our previous Delphi study indicated that rehabilitation experts rated “group rehabilitation” as neither effective nor ineffective in motivating these individuals³⁰. On the basis of these findings, we hypothesized that patients and clinicians would differ to some extent in their perceptions of the relative importance of factors that motivate patients for rehabilitation. However, to the best of our knowledge, no studies have directly compared patients’ preferences regarding motivational factors with those of clinicians. Therefore, the current study aimed to compare patients’ and clinicians’ perceptions of the most important factors in motivating patients for rehabilitation.”

In addition, we have added the following references.

- 2 Maclean, N., Pound, P., Wolfe, C. & Rudd, A. Qualitative analysis of stroke patients' motivation for rehabilitation. *BMJ* **321**, 1051-1054 (2000).
- 3 Maclean, N., Pound, P., Wolfe, C. & Rudd, A. The concept of patient motivation: A qualitative analysis of stroke professionals' attitudes. *Stroke* **33**, 444-448 (2002).
- 14 Kim, S. I. Neuroscientific model of motivational process. *Front. Psychol.* **4**, 98; 10.3389/fpsyg.2013.00098 (2013).
- 15 Pessiglione, M., Vinckier, F., Bouret, S., Daunizeau, J. & Le Bouc, R. Why not try harder? Computational approach to motivation deficits in neuro-psychiatric diseases. *Brain* **141**, 629-650 (2018).
- 16 Simpson, E. H. & Balsam, P. D. The behavioral neuroscience of motivation: An overview of concepts, measures, and translational applications. *Curr. Top. Behav. Neurosci.* **27**, 1-12 (2016).
- 17 Vu, T. *et al.* Motivation-achievement cycles in learning: A literature review and research agenda. *Educ. Psychol. Rev.* **34**, 39-71 (2022).
- 18 Ryan, T. A. *Intentional behavior: An approach to human motivation.* (Ronald Press, 1970).
- 19 Raynor, J. O. & McFarlin, D. B. Foundations of social behavior in *Handbook of motivation and cognition* (eds. Sorrentino, R.M. & Higgins, E.T.) 315-349 (Guilford Press, 1986).
- 20 Deci, E. L. & Ryan, R. M. Self-determination theory in health care and its relations to motivational interviewing: A few comments. *Int. J. Behav. Nutr. Phys. Act.* **9**, 24; 10.1186/1479-5868-9-24 (2012).
- 21 Schunk, D. H. Self-efficacy and academic motivation. *Educ. Psychol.* **26**, 207-231 (1991).
- 22 Wentzel, K. R. Social relationships and motivation in middle school: The role of parents, teachers, and peers. *J. Educ. Psychol.* **90**, 202-209 (1998).
- 23 Tamura, A. *et al.* Exploring the within-person contemporaneous network of motivational engagement. *Learn. Instr.* **81**, 101649; 10.1016/j.learninstruc.2022.101649 (2022).
- 24 Murayama, K. & Elliot, A. J. The competition-performance relation: A meta-analytic review and test of the opposing processes model of competition and performance. *Psychol. Bull.* **138**, 1035-1070 (2012).
- 25 Mead, N. & Bower, P. Patient-centredness: A conceptual framework and review of the empirical literature. *Soc. Sci. Med.* **51**, 1087-1110 (2000).
- 26 Gartner, J. B. *et al.* Definition and conceptualization of the patient-centered care pathway, a proposed integrative framework for consensus: A concept

analysis and systematic review. *BMC Health Serv. Res.* **22**, 558; 10.1186/s12913-022-07960-0 (2022).

● **Comment 2**

Methodology - I wouldn't describe this as a cross sectional study instead consider this survey research. There are aspects that are explanatory survey research.

Response

In accordance with your comment, we have revised the following sentences in the Title, Abstract, and Methods sections.

Title (page 1, line 2)

“Patients’ and Clinicians’ Perceptions of Motivational Factors in Rehabilitation: A Multicenter Explanatory Survey”

Abstract (page 1, line 9)

“Methods: This multicenter explanatory survey research was conducted from January to March 2022.”

Methods (page 5, line 76)

“We employed a multicenter explanatory survey research design.”

● **Comment 3**

Data analysis - The odds ratio doesn't fit well for me. Because I don't see the distinction between patient and therapist as 'exposure' I don't see the fit for the odds ratio in the primary analysis. I would like a strong justification for the choice of this method. The introduction hypothesized that there was a difference between clinicians and patients – is the odds ratio the best way to show there is a difference? The results section state the perceptions are similar but don't provide the statistics to fully support those statements. I see the value of odds ratio related to patient characteristics and your regression analysis.

Comment 3-1: *The odds ratio doesn't fit well for me. Because I don't see the distinction between patient and therapist as 'exposure' I don't see the fit for the odds ratio in the primary analysis. I would like a strong justification for the choice of this method. The introduction hypothesized that there was a difference between clinicians and patients – is the odds ratio the best way to show there is a difference?*

Response

We thank the reviewer for pointing out our misunderstanding. We compared patients’ responses with those of clinicians using Fisher’s exact test in the previous manuscript. Although we also used the odds ratio as the measure of effect size in the previous manuscript, after receiving your comment, we have reconsidered this method and have used phi coefficients instead of odds ratios. We have added the following sentences to the revised Methods section to describe our data analyses more clearly.

Page 10, line 157 to page, 10, line 160

“We compared patients’ responses with those of clinicians using Fisher’s exact test. Phi coefficients with 95% CIs were calculated as the measure of effect size for comparing responses between groups.”

We have also added p-values of Fisher’s exact test, and phi coefficients with 95% confidence intervals in the Abstract and Results sections, as follows.

Abstract (page 2, lines 19–23)

“Of these nine motivational factors, “medical information” ($p < 0.001$; $\phi = -0.14$; 95% confidence interval = -0.20 to -0.07) and “control of task difficulty” ($p = 0.011$; $\phi = -0.09$; 95% confidence interval = -0.16 to -0.02) were selected by a significantly higher proportion of patients than clinicians.”

Results (page 11, line 191 to page 12, line 199)

“The phi coefficients comparing patients’ top choices with those of clinicians are shown in Figure 2. Of the nine motivational factors selected by more than 5% of patients, “medical information” ($p < 0.001$; $\phi = -0.14$; 95% CI = -0.20 to -0.07) and “control of task difficulty” ($p = 0.011$; $\phi = -0.09$; 95% CI = -0.16 to -0.02) were chosen by a significantly higher proportion of patients than clinicians. In contrast, a significantly higher proportion of clinicians than patients rated “realization of recovery” ($p = 0.001$; $\phi = 0.11$; 95% CI = 0.04 to 0.18) and “goal setting” ($p = 0.005$; $\phi = 0.10$; 95% CI = 0.03 to 0.16) as the most important.”

Results (page 13, lines 213–225)

“The phi coefficients comparing patients’ choices with those of clinicians for the first question of the survey are shown in Figure 4. A significantly higher proportion of patients than clinicians rated “a suitable rehabilitation environment” ($p < 0.001$; $\phi = -0.19$; 95% CI = -0.26 to -0.12), “rehabilitation programs with variations” ($p < 0.001$; $\phi = -0.16$; 95% CI = -0.23 to -0.09), “control of task difficulty” ($p < 0.001$; $\phi = -0.15$; 95% CI = -0.22 to -0.09), “respect for self-determination” ($p < 0.001$; $\phi = -0.14$; 95% CI = -0.21 to -0.07), and “medical information” ($p < 0.001$; $\phi = -0.14$; 95% CI = -0.20 to -0.07) as important. Conversely, “presence of family members during rehabilitation” ($p = 0.002$; $\phi = 0.19$; 95% CI = 0.12 to 0.26), “goal setting” ($p < 0.001$; $\phi = 0.18$; 95% CI = 0.12 to 0.25), “enjoyable rehabilitation programs” ($p < 0.001$; $\phi = 0.13$; 95% CI = 0.06 to 0.19), and “realization of recovery” ($p < 0.001$; $\phi = 0.12$; 95% CI = 0.05 to 0.19) were chosen by a significantly higher proportion of clinicians than patients.”

Furthermore, we have revised the keywords, Figures 2 and 4, and the legends of Figures 2 and 4, as follows.

Keywords (page 2, line 31)

“**Keywords:** Motivation, Patient-centered care, Phi coefficient, Strategy”

The legend of Figure 2 (page 34, lines 581–583)

“**Figure 2. Comparison of patients’ and clinicians’ perceptions of the most important motivational factors.** Motivational factors are arranged in ascending order regarding the value of the phi coefficient. Filled diamonds and error bars represent phi coefficients and 95% confidence intervals, respectively. Values are

number (%) unless indicated.”

The legend of Figure 4 (page 35, lines 591–594)

“**Figure 4. Comparison of patients’ and clinicians’ perceptions of the three most important motivational factors.** Motivational factors are arranged in ascending order regarding the value of the **phi coefficient**. Filled diamonds and error bars represent **phi coefficients** and 95% confidence intervals, respectively. **Values are number (%) unless indicated.**”

Please see the revised Figures 2 and 4 for details.

***Comment 3-2:** The results section state the perceptions are similar but don't provide the statistics to fully support those statements.*

Response

Our statement that patients’ and clinicians’ perceptions were similar was based on the results of *descriptive statistics* showing that the three most frequently selected motivational factors were identical for the two groups. This study had a large sample size determined using a priori sample size calculation and sufficient response rates. These procedures would be expected to strengthen the reliability of our descriptive statistics results.

Nevertheless, we agree with the reviewer’s comment, and have conducted the following supplemental analysis to assess the reliability of the results. First, we randomly sampled approximately half of the participants (240 patients and 200 clinicians) 10 times with replacement from all participants. Second, we computed the mean and 95% confidence interval for the percentage of participants who selected each potential motivational factor. The analysis revealed that “realization of recovery,” “goal setting,” and “practice related to the patient’s experience and lifestyle” were also most frequently selected by both patients and clinicians, which supports the reliability of our descriptive statistics results.

We have added the following sentence to the Discussion section to state that the results of the supplemental analysis support the reliability of the results of descriptive analysis.

Page 15, lines 261–263

“The first novel finding regarding similarity in perceptions **was** that not only clinicians, but also patients **considered** “goal setting” and “practice related to the patient’s experience and lifestyle” as **the** most important **factor**. **The results of our supplemental analysis, obtained by repeated random sampling with replacement from all participants, also support the reliability of this finding (Supplemental Figures S1 and S2).**”

We have also added Supplemental Figures S1 and S2 to the Supplementary material to show the results of the supplemental analysis.

The newly added Supplemental Figures S1 and S2 are shown below.

a

The distribution of patients' answer (n = 240)

b

The distribution of clinicians' answer (n = 200)

Figure S1. Distributions of patients' (a) and clinicians' (b) answers regarding the most important motivational factor obtained by randomly sampling approximately half of the participants (240 patients and 200 clinicians) 10 times with replacement from all participants.

Motivational factors are arranged in descending order by the mean of the percentage of participants. Error bars represent 95% confidence intervals. The vertical dashed line represents 5% of participants.

a

The distribution of patients' answer (n = 240)

b

The distribution of clinicians' answer (n = 200)

Figure 2. Distribution of patients' (a) and clinicians' (b) answers regarding the three most important motivational factors obtained by repeated random sampling with replacement from all participants.

Motivational factors are arranged in descending order by the mean of the percentage of participants. Error bars represent 95% confidence intervals. The vertical dashed line represents 5% of participants.

Comment 3-3: *I see the value of odds ratio related to patient characteristics and your regression analysis.*

Response

Thank you for your positive evaluation of the multiple logistic regression analysis in our manuscript. We have added a more detailed explanation of the results of the regression analyses in the Results section according to Reviewer #2's comment 4-2. Please see our response to Reviewer #2's comment 4-2.

● **Comment 4**

Results - some of the content is better placed in discussion. For example line 239 citation of other work would be better in a discussion where you can compare and contrast your results with others. As the reader I would benefit from a stronger focus on the statistics you obtained here. I want the tables to be fully explained.

Comment 4-1: *Results - some of the content is better placed in discussion. For example line 239 citation of other work would be better in a discussion where you can compare and contrast your results with others.*

Response

We apologize for the confusion regarding this point. In the previous version of our manuscript, we had described the comparisons of our results with those of previous studies (including the statement on Line 239) in the Discussion section, but not in the Results section. We have added text to the Discussion section of the revised manuscript to address this issue.

Comment 4-2: *As the reader I would benefit from a stronger focus on the statistics you obtained here. I want the tables to be fully explained.*

Response

Thank you for your helpful comment. We have added further explanations of the results of statistical analyses and the Tables to the Results section, as follows.

Page 11, line 193 to page 12, line 199

“Of the nine motivational factors selected by more than 5% of patients, “medical information” ($p < 0.001$; $\phi = -0.14$; 95% CI = -0.20 to -0.07) and “control of task difficulty” ($p = 0.011$; $\phi = -0.09$; 95% CI = -0.16 to -0.02) were chosen by a significantly higher proportion of patients than clinicians. **In contrast, a significantly higher proportion of clinicians than patients rated “realization of recovery” ($p = 0.001$; $\phi = 0.11$; 95% CI = 0.04 to 0.18) and “goal setting” ($p = 0.005$; $\phi = 0.10$; 95% CI = 0.03 to 0.16) as the most important.”**

Page 13, lines 214–225

“A significantly higher proportion of patients than clinicians rated “a suitable rehabilitation environment” ($p < 0.001$; $\phi = -0.19$; 95% CI = -0.26 to -0.12), “rehabilitation programs with variations” ($p < 0.001$; $\phi = -0.16$; 95% CI = -0.23 to -0.09), “control of task difficulty” ($p < 0.001$; $\phi = -0.15$; 95% CI = -0.22 to -0.09), “respect for self-determination” ($p < 0.001$; $\phi = -0.14$; 95% CI = -0.21 to -0.07), and “medical information” ($p < 0.001$; $\phi = -0.14$; 95% CI

= -0.20 to -0.07) as important. Conversely, “presence of family members during rehabilitation” (p = 0.002; phi = 0.19; 95% CI = 0.12 to 0.26), “goal setting” (p < 0.001; phi = 0.18; 95% CI: 0.12 to 0.25), “enjoyable rehabilitation programs” (p < 0.001; phi = 0.13; 95% CI = 0.06 to 0.19), and “realization of recovery” (p < 0.001; phi = 0.12; 95% CI = 0.05 to 0.19) were chosen by a significantly higher proportion of clinicians than patients.”

Page 14, lines 232–243

“Most patients were those with stroke (n = 218; 51.9%), female (n = 240; 57.1%), and aged ≥ 65 years (n = 352; 84.3%). The median length of hospital stay was 43 days (interquartile range, 34 to 64 days). The results of the multiple logistic regression analysis are shown in Table 3. A significantly higher proportion of patients aged < 65 years old (n = 24; 36.2%), compared with those aged ≥ 65 years old (n = 84; 23.8%), rated “realization of recovery” (odds ratio = 0.53; 95% CI = 0.29 to 0.97) as the most important factor. In addition, “goal setting” (odds ratio = 0.46; 95% CI = 0.22 to 0.93) was also chosen by a significantly larger proportion of patients aged < 65 years old (n = 15; 22.7%) compared with those aged ≥ 65 years old (n = 50; 8.5%). Furthermore, patients with shorter hospital stay lengths were significantly more likely to choose “medical information” as the most important factor (odds ratio = 0.97; 95% CI = 0.94 to 1.00), although only 27 of 420 patients chose it.”

● **Comment 5**

224-231 do not belong in discussion section.

Response

In accordance with your recommendation, we have removed the paragraph that you pointed out.

● **Comment 6**

283-288- need to either better support or cite statements like this.

Response

We appreciate the reviewer’s helpful suggestions and have added the findings of previous studies that support our results to the Discussion section, as follows.

Page 18, line 309 to page 19, line 321

“Goal setting has been shown to be a more important motivator for physical activity in younger people compared with older people⁶⁴. In our previous qualitative study of physical therapists, participants listed setting goals, such as returning to work and society, as an effective motivational strategy for relatively young patients³⁶. In addition, younger people with physical disabilities tend to have higher expectations of what they can achieve, such as wanting to be able to participate in sporting activities or to lead an active social life, compared with other age groups⁶⁵. Furthermore, a qualitative study with patients with stroke in the intensive inpatient rehabilitation ward suggested that improvement in physical function has a more positive effect on motivation among relatively younger compared with older patients⁴⁷. These previous findings support the results of the present study. Therefore, setting goals that are valuable to patients and helping

them experience positive achievement emotions may be especially important for enhancing active participation in rehabilitation for relatively young patients.”

We have also added the following references.

64 Spiteri, K. *et al.* Barriers and motivators of physical activity participation in middle-aged and older-adults – a systematic review. *J. Aging Phys. Act.* **27**, 929-944 (2019).

65 Evans, S. A. *et al.* Disability in young adults following major trauma: 5 year follow up of survivors. *BMC Public Health* **3**, 8; 10.1186/1471-2458-3-8 (2003).

● **Comment 7**

Is Table 1 the actual survey that was given? Did the participants have these examples? Provide the actual survey questions that were read to patients the written survey given to the clinicians.

Comment 7-1: *Is Table 1 the actual survey that was given? Did the participants have these examples?*

Response

The list of potential motivational factors with specific examples shown in Table 1 was presented to clinicians, when they responded to the survey. In contrast, patients were presented with the list without specific examples, because they could receive verbal explanations of the specific examples from the interviewer. To describe these points clearly, we have added the following sentences to the Methods section.

Page 8, lines 124–126

“The list presented to patients did not include the specific examples shown in Table 1 (Supplemental Box S1), because they received verbal explanations of specific examples from the interviewer.”

Page 9, line 145

“In the first question, clinicians were asked to select the three most important factors for increasing patient adherence to rehabilitation programs from the list shown in Table 1.”

We have added the list presented to patients as the Supplemental Box S1 to the Supplementary material. Please see the newly added Supplemental Box S1 for details.

In addition, we have revised the Title of Table 1 to clarify that Table 1 shows the list presented to clinicians, as follows.

Title of Table 1 (Page 36, line 595)

“**Table 1. List of potential motivational factors for clinicians**”

Comment 7-2: *Provide the actual survey questions that were read to patients the*

written survey given to the clinicians.

Response

In accordance with your suggestion, we have added the structured interview guide used for interviews with patients as Supplemental Box S2 and the actual survey questions presented to clinicians as Supplemental Box S3 to the Supplementary material.

We have added the following sentences in the Methods section to state the above clearly.

Page 8, lines 131–132

“The structured interview included two questions. In the first question, patients were asked to select the three most important factors for facilitating their engagement in rehabilitation from the list. In the second question, they were instructed to choose the most important factor from the three factors they selected in the first question. Participants who selected “other” were asked to respond to an open-ended question in which they proposed additional motivational factors.

The structured interview guide is shown in Supplemental Box S2.”

Page 9, lines 142–143

“The survey questions that were given to clinicians are shown in Supplemental Box S3.”

Please see the newly added Supplemental Boxes 2 and 3 for details.

● **Comment 8**

Table 3 - there were very limited sig associations. Why do you think this is? Is this a problem in the model?

Response

A small sample size may explain why the combinations of patients’ choices and their demographic characteristics that showed statistically significant associations were limited. The number of patients who selected the 12 motivational factors that were not significantly associated with the demographic characteristics of patients was less than 50. The small sample size results in large confidence intervals for the odds ratio, which has the potential to reduce the power to detect statistically significant associations between patients’ choices and their demographic characteristics.

We have added the following sentences in the Discussion section to state this point clearly.

Page 19, lines 328–335

“The limited combinations of patients’ choices and their demographic characteristics that showed statistically significant associations may be explained by a small sample size. The number of patients who selected the 12 motivational factors not significantly associated with any of the demographic variables was less than 50, which resulted in large CIs for the odds ratio. Thus, as the small sample size could reduce the power to detect statistically significant associations between patients’ choices and their demographic characteristics, careful interpretation of the results of multiple logistic regression analyses is necessary.”

- **Comment 9**

Line 171 I'm not clear what those ranges mean. I would like to see the full data in a table.

Response

Thank you for pointing out this issue. The ranges of percentages indicate the percentages of patients and clinicians who chose “realization of recovery,” “goal setting,” and “practice related to the patient’s experience and lifestyle.” To clarify the percentages of participants who chose these factors, we have revised the relevant statement in the Results section, as follows.

Page 11, lines 183–188

“The three most frequently selected motivational factors were identical for patients and clinicians: “realization of recovery” chosen by 26.5% of patients and 36.7% of clinicians, “goal setting” chosen by 15.0% of patients and 22.4% of clinicians, and “practice related to the patient’s experience and lifestyle” chosen by 10.4% of patients and 9.5% of clinicians.”

Page 12, lines 207–212

“Similar to the results regarding the motivational factors perceived as the most important, the three most frequently endorsed motivational factors were identical for patients and clinicians: “realization of recovery” chosen by 47.4% of patients and 59.4% of clinicians, “goal setting” chosen by 34.7% of patients and 52.6% of clinicians, and “practice related to the patient’s experience and lifestyle” chosen by 32.4% of patients and 38.2% of clinicians.”

In addition, according to your comment, we have added the percentages of participants who selected the relevant item to Figures 2 and 4. Please see the revised Figures 2 and 4.

We have also changed the legends of Figures 2 and 4, as follows.

Page 34, lines 581–583

“**Figure 2. Comparison of patients’ and clinicians’ perceptions of the most important motivational factors.** Motivational factors are arranged in ascending order regarding the value of the phi coefficient. Filled diamonds and error bars represent phi coefficients and 95% confidence intervals, respectively. Values are number (%) unless indicated.”

Page 35, lines 591–594

“**Figure 4. Comparison of patients’ and clinicians’ perceptions of the three most important motivational factors.** Motivational factors are arranged in ascending order regarding the value of the phi coefficient. Filled diamonds and error bars represent phi coefficients and 95% confidence intervals, respectively. Values are number (%) unless indicated.”

Reviewers' comments:

Reviewer #1 (Remarks to the Author):

The authors have addressed all the concerns and questions mentioned by the reviewers. As a minor comment, it might be nice to provide some examples of the different perceptions between patients and clinicians entitled in lines 61-63. Were there any important finding or missing knowledge outlined in reference 29 that would have been considered in planning the present study? It would be helpful to rephrase such, if available.

Lines 157-160 more information on the selected statistics would be helpful, e.g. how is the Phi coefficient defined (including an equation and which variables of the Fisher's exact test were used, if applicable) and interpreted in terms of the effect sizes and the comparison made (clinicians vs. patients perspectives?).

Apart from these minor revisions, the revised manuscript adds relevant information for the research and general community and can be recommended publication.

Reviewer #3 (Remarks to the Author):

The manuscript has already been reviewed by two reviews and has been soundly revised. The introduction has been expanded, more detail regarding the analysis is included in the results section and the discussion section has been clarified.

For me, as a reader, further information on the context of the study would still be helpful. My concrete questions are:

How are motivational factors included in the concept of the wards? Are there any guidelines? Is there any training for health care professionals with regard to motivational factors? Does

You used a questionnaire with 15 motivational factors that was developed in your context – do you find these factors in the international literature?

Finally, but this is just a question of interest: Are those patients who reported unfrequent/less important motivational factors special according to sociodemographic characteristics?

Responses to the reviewers' comments

To Reviewer #1:

We thank the reviewer for the time devoted to reviewing our manuscript and for the helpful comments. We have addressed each comment in a point-by-point manner below. In addition, we have used a native English proofreading service again to refine our manuscript and have made some corrections in the manuscript in accordance with the proofreader's suggestions. Please see the revised manuscript for details.

● **Comment 1**

As a minor comment, it might be nice to provide some examples of the different perceptions between patients and clinicians entitled in lines 61-63. Were there any important finding or missing knowledge outlined in reference 29 that would have been considered in planning the present study? It would be helpful to rephrase such, if available.

Response

We appreciate the reviewer's helpful comment. Plant et al. (reference 29) reported that patients and clinicians differ in their perspectives of goal setting, which is one of the key motivational factors. Specifically, the patients' focus was on the long-term, regaining physical function and independence, and returning to former activities and roles. In contrast, the clinicians' goals tended to be short-term, specific, conservative in ambition, and driven by financial and organizational pressures. These are important findings, indicating that there may be differences in the perceptions of the relative importance of motivational factors between patients and clinicians.

We have added the following sentences to the Introduction section to provide examples of the different perceptions between patients and clinicians.

Page 5, lines 70–74

“An example of this hinderance is that, regarding goal setting, patients appear to focus on the long-term, regaining physical function and independence and returning to former activities and roles. In contrast, clinicians' goals tend to be short-term, specific, conservative in ambition, and driven by financial and organizational pressure²⁹. Additionally, with regard to rehabilitation programs, group exercise has been reported to be a perceived motivator of physical activity for individuals with stroke⁴. However, our previous Delphi study indicated that rehabilitation experts rated group rehabilitation as neither effective nor ineffective in motivating these individuals³⁰.”

● **Comment 2**

Lines 157-160 more information on the selected statistics would be helpful, e.g. how is the Phi coefficient defined (including an equation and which variables of the Fisher's exact test were used, if applicable) and interpreted in terms of the effect sizes and the comparison made (clinicians vs. patients perspectives?).

Response

We thank the reviewer for the comment. We have added the following sentences to the Statistics and reproducibility section to provide more information on the Phi coefficient.

Page 11, line 175 to page 12, line 183

“Phi coefficients with 95% CIs were calculated as the measure of effect size for comparing responses between groups with the following equation:

$$\text{Phi coefficient} = (A \times D - B \times C) / \{(A + B) \times (C + D) \times (A + C) \times (B + D)\}^{1/2}$$

where A is the number of clinicians who selected a motivational factor, B is the number of clinicians who did not select this factor, C is the number of patients who selected this factor, and D is the number of patients who did not select this factor⁴⁹. Negative values of the phi coefficient indicate that a higher proportion of patients than clinicians rated the factor as important/most important. An absolute value of a phi coefficient of 0.05 was considered as a weak effect size, 0.10 a moderate effect size, 0.15 a strong effect size, and 0.25 a very strong effect size⁵⁰.”

In addition, we have added the following references.

49 Metsämuuronen, J. Artificial systematic attenuation in eta squared and some related consequences: attenuation-corrected eta and eta squared, negative values of eta, and their relation to Pearson correlation. *Behaviormetrika* **50**, 27-61 (2023).

50 Akoglu, H. User's guide to correlation coefficients. *Turk. J. Emerg. Med.* **18**, 91-93 (2018).

To Reviewer #3:

We thank the reviewer for the time spent in reviewing our manuscript and the helpful comments. We have addressed each comment in a point-by-point manner below. Additionally, we have used a native English proofreading service again to refine our manuscript and have revised some parts of the manuscript following the proofreader's suggestions. Please see the revised manuscript for details.

● **Comment 1**

For me, as a reader, further information on the context of the study would still be helpful. My concrete questions are: How are motivational factors included in the concept of the wards? Are there any guidelines? Is there any training for health care professionals with regard to motivational factors? Does you used a questionnaire with 15 motivational factors that was developed in your context – do you find these factors in the international literature?

Comment 1-1: *How are motivational factors included in the concept of the wards?*

Response

We thank the reviewer for the helpful comment. Although motivational factors are not explicitly included in the concept/goals of the intensive inpatient rehabilitation ward, these are elements of good clinical practice in rehabilitation. In particular, goal setting is an essential procedure to facilitate an interdisciplinary team approach in the wards (Miyai et al., *Neurorehabil Neural Repair*. 2011). We have added the following sentence to the Discussion section to provide an example that goal setting is an essential procedure not only for motivating patients but also for good clinical practice.

Page 17, lines 285–287

“Previous theoretical, experimental, and observational studies have reported that these two factors are essential components of motivation^{11,30,35,36,53-55}. **In addition, goal setting is regarded as one of the important procedures to facilitate an interdisciplinary team approach³¹.** Therefore, many clinicians use these factors as key-motivational strategies in intensive inpatient rehabilitation wards.”

Comment 1-2: *Are there any guidelines?*

Response

To the best of our knowledge, there are no guidelines on motivational factors in rehabilitation. Therefore, we believe that the findings of this study provide rehabilitation clinicians with helpful information for effectively motivating patients to engage in rehabilitation.

Comment 1-3: *Is there any training for health care professionals with regard to motivational factors?*

Response

No, there is no training for health care professionals regarding motivational factors. We previously reported that many rehabilitation clinicians do not undergo sufficient academic training in motivational strategies and acquire skills to motivate patients via clinical experience (Oyake et al., *Rehabilitation Kyouiku*

Kenkyu. 2021; Frontiers in Neurology, 2020). Therefore, we have added the following sentence to the Discussion section to emphasize the clinical significance of our findings.

Page 17, lines 275–277

“Additionally, because there are no guidelines and adequate training programs for clinicians regarding motivational strategies^{35,52}, our findings may provide clinicians with helpful information for effectively motivating patients to engage in rehabilitation.”

We have also added the following reference.

52 Oyake, K. & Tanaka, S. Opportunity where medical professionals learn motivational strategies for stroke rehabilitation: A descriptive cross-sectional study. *Rehabilitation Kyouiku Kenkyu* 27, 8-13 (2021) (in Japanese).

Comment 1-4: *Does you used a questionnaire with 15 motivational factors that was developed in your context – do you find these factors in the international literature?*

Response

We thank the reviewer for this question. Yes, these factors are reported in the international literature. We developed our list of potential motivational factors on the basis of findings from our previous studies and the international literature. Therefore, as mentioned in the Discussion section (page 21, lines 353–355), our results support the results of these studies conducted in different countries. We have added the following sentence to the Methods section to clarify this point.

Page 8, lines 126–128

“The first author initially developed a list of potential factors involved in increasing the patients’ motivation for rehabilitation on the basis of findings from our previous studies^{30,35,36} and related international literature^{3-6,9,37-46}.”

● **Comment 2**

Finally, but this is just a question of interest: Are those patients who reported unfrequent/less important motivational factors special according to sociodemographic characteristics?

Response

We appreciate the reviewer’s question. However, regarding 12 motivational factors that were rated as the most important by less than 50 (10.4%) patients, we could not detect any statistically significant differences in demographic characteristics between patients who selected these factors and those who did not. As noted in the Discussion section (from page 20, line 344 to page 21, line 351), these results may be explained by the small sample size. Therefore, further studies with a large sample size are required to investigate this issue.

REVIEWERS' COMMENTS:

Reviewer #1 (Remarks to the Author):

Thanks to the authors for addressing all of the reviewers queries. The manuscript can be recommended for publication.

Reviewer #3 (Remarks to the Author):

Thank you for the revision according to the reviewers' comments. The comments have been addressed although some could not be incorporated into the manuscripts. Reasons are given.

The manuscripts expands the knowledge on motivational factors in rehabilitation. I recommend the manuscript for publication.